# Matching on Balanced Nonlinear Representations for Treatment Effects Estimation

**Sheng Li**
Adobe Research
San Jose, CA
sheli@adobe.com

**Yun Fu**
Northeastern University
Boston, MA
yunfu@ece.neu.edu

## Abstract

Estimating treatment effects from observational data is challenging due to the missing counterfactuals. Matching is an effective strategy to tackle this problem. The widely used matching estimators such as nearest neighbor matching (NNM) pair the treated units with the most similar control units in terms of covariates, and then estimate treatment effects accordingly. However, the existing matching estimators have poor performance when the distributions of control and treatment groups are unbalanced. Moreover, theoretical analysis suggests that the bias of causal effect estimation would increase with the dimension of covariates. In this paper, we aim to address these problems by learning low-dimensional balanced and nonlinear representations (BNR) for observational data. In particular, we convert counterfactual prediction as a classification problem, develop a kernel learning model with domain adaptation constraint, and design a novel matching estimator. The dimension of covariates will be significantly reduced after projecting data to a low-dimensional subspace. Experiments on several synthetic and real-world datasets demonstrate the effectiveness of our approach.

## 1 Introduction

Causal questions exist in many areas, such as health care [24, 12], economics [14], political science [17], education [36], digital marketing [6, 43, 5, 15, 44], etc. In the field of health care, it is critical to understand if a new medicine could cure a certain illness and perform better than the old ones. In political science, it is of great importance to evaluate whether the government should fund a job training program, by assessing if the program is the true factor that leads to the success of job hunting. All of these causal questions can be addressed by the causal inference technique. Formally, causal inference estimates the treatment effect on some units after interventions [33, 20]. In the above example of heath care, the units could be patients, and the intervention would be taking new medicines. Due to the wide applications of causal questions, effective causal inference techniques are highly desired to address these problems.

Generally, the causal inference problems can be tackled by either experimental study or observational study. Experimental study is popular in traditional causal inference problems, but it is time-consuming and sometimes impractical. As an alternative strategy, observational study has attracted increasing attention in the past decades, which extracts causal knowledge only from the observed data. Two major paradigms for observational study have been developed in computer science and statistics, including the causal graphical model [29] and the potential outcome framework [27, 33]. The former builds directed acyclic graphs (DAG) from covariates, treatment and outcome, and uses probabilistic inference to determine causal relationships; while the latter estimates counterfactuals for each treated unit, and gives a precise definition of causal effect. The equivalence of two paradigms has been discussed in [11]. In this paper, we mainly focus on the potential outcome framework.

A *missing data problem* needs to be dealt with in the potential outcome framework. As each unit is either treated or not treated, it is impossible to observe its outcomes in both scenarios. In other words, one has to predict the *missing counterfactuals*. A widely used solution to estimating counterfactuals is matching. According to the (binary) treatment assignments, a set of units can be divided into a treatment group and a control group. For each treated unit, matching methods select its counterpart in the control group based on certain criteria, and treat the selected unit as a counterfactual. Then the treatment effect can be estimated by comparing the outcomes of treated units and the corresponding counterfactuals. Some popular matching estimators include nearest neighbor matching (NNM) [32], propensity score matching [31], coarsened exact matching (CEM) [17], genetic matching [9], etc.

Existing matching methods have three major drawbacks. First, they either perform matching in the original covariate space (e.g., NNM, CEM) or in the one-dimensional propensity score space (e.g., PSM). The potential of using intermediate representations has not been extensively studied before. Second, existing methods work well for data with a moderate number of covariates, but may fail for data with a large number of covariates, as theoretical analysis suggests that the bias of treatment effect estimation would increase with the dimension of covariates [1]. Third, most matching methods do not take into account whether the distributions of two groups are balanced or not. The matching process would make no sense if the distributions of two groups have little overlap.

To address the above problems, we propose to learn balanced and nonlinear representations (BNR) from observational data, and design a novel matching estimator named BNR-NNM. First, the counterfactual prediction problem is converted to a multi-class classification problem, by categorizing the outcomes to ordinal labels. Then, we propose a novel criterion named ordinal scatter discrepancy (OSD) for supervised kernel learning on data with ordinal labels, and extract low-dimensional nonlinear representations from covariates. Further, to achieve balanced distributions in the low-dimensional space, a maximum mean discrepancy (MMD) criterion [4] is incorporated to the model. Finally, matching strategy is performed on the extracted balanced representations, in order to provide a robust estimation of causal effect. In summary, the main contributions of our work include:

- We propose a novel matching estimator, BNR-NNM, which learns low-dimensional balanced and nonlinear representations via kernel learning.
- We convert the counterfactual prediction problem into a multi-class classification problem, and design an OSD criterion for nonlinear kernel learning with ordinal labels.
- We incorporate a domain adaptation constraint to feature learning by using the maximum mean discrepancy criterion, which leads to balanced representations.
- We evaluate the proposed estimator on both synthetic datasets and real-world datasets, and demonstrate its superiority over the state-of-the-art methods.

## 2 Background

**Potential Outcome Framework.** The potential outcome framework is proposed by Neyman and Rubin [27, 33]. Considering binary treatments for a set of units, there are two possible outcomes for each unit. Formally, for unit $k$, the outcome is defined as $Y_k(1)$ if it received treatment, and $Y_k(0)$ if it did not. Then, the individual-level treatment effect is defined as $\gamma_k = Y_k(1) - Y_k(0)$. Clearly, each unit only belongs to one of the two groups, and therefore, we can only observe one of the two possible outcomes. This is the well-known *missing data problem* in causal inference. In particular, if unit $k$ received treatment, $Y_k(1)$ is the observed outcome, and $Y_k(0)$ is missing data, i.e., *counterfactual*.

The potential outcome framework usually makes the following assumptions [19].

**Assumption 1.** *Stable Unit Treatment Value Assumption (SUTVA): The potential outcomes for any units do not vary with the treatments assigned to other units, and for each unit there are no differences forms or versions of each treatment level, which lead to different potential outcomes.*

**Assumption 2.** *Strongly Ignorable Treatment Assignment (SITA): Conditional on covariates $x_k$, treatment $T_k$ is independent of potential outcomes.*

$$(Y_k(1), Y_k(0)) \perp\!\!\!\perp T_k | x_k. \quad \text{(Unconfoundedness)}$$
$$0 < \Pr(T_k = 1 | x_k) < 1. \quad \text{(Overlap)} \tag{1}$$

These assumptions enable the modeling of treatment of one unit with respect to covariates, independent of outcomes and other units.

**Matching Estimators.** To address the aforementioned missing data problem, a simple yet effective strategy has been developed, which is *matching* [32, 33, 14, 40]. The idea of matching is to estimate

the counterfactual for a treated unit by seeking its most similar counterpart in the control group. Existing matching methods can be roughly divided into three categories: nearest neighbor matching (NNM), weighting, and subclassification. We mainly focus on NNM in this paper.

Let $X_{\mathrm{C}} \in \mathbb{R}^{d \times N_C}$ and $X_{\mathrm{T}} \in \mathbb{R}^{d \times N_T}$ denote the covariates of a control group and a treatment group, respectively, where $d$ is the number of covariates, $N_C$ and $N_T$ are the group sizes. $T$ is a binary vector indicating if the units received treatments (i.e., $T_k = 1$) or not (i.e., $T_k = 0$). $Y$ is an outcome vector. For each treated unit $k$, NNM finds its nearest neighbor in the control group in terms of the covariates. The outcome of the selected control unit is considered as an estimation of counterfactual. Then, the average treatment effect on treated (ATT) is defined as:

$$ATT = \frac{1}{N_T} \sum_{k:T_k=1} \left( Y_k(1) - \hat{Y}_k(0) \right), \tag{2}$$

where $\hat{Y}_k(0)$ is the counterfactual estimated from unit $k$'s nearest neighbor in the control group.

NNM can be implemented in various ways, such as using different distance metrics, or choosing different number of neighbors. Euclidean distance and Mahalanobis distance are two widely-used distance metrics for NNM. They work well when there are a few covariates with normal distributions [34]. Another important matching estimator is propensity score matching (PSM) [31]. PSM estimates the propensity score (i..e., the probability of receiving treatment) for each unit via logistic regression, and pairs the units from two groups with similar scores [35, 8, 30]. Most recently, a covariate balancing propensity score (CBPS) method is developed to balance the distributions of two groups by weighting the covariates, and has shown promising performance [18].

The key differences between the proposed BNR-NNM estimator and the traditional matching estimators are two-fold. First, BNR-NNM performs matching in an intermediate low-dimensional subspace that could guarantee a low estimation bias, while the traditional estimators adopt either the original covariate space or the one-dimensional space. Second, BNR-NNM explicitly considers the balanced distributions across treatment and control groups, while the traditional estimators usually fail to achieve such a property.

**Machine Learning for Causal Inference.** In recent years, researchers have been exploring the relationships between causal inference and machine learning [39, 10, 38]. A number of predictive models have been designed to estimate the causal effects, such as causal trees [3] and causal forests [42]. Balancing the distributions of two groups is considered as a key issue in observational study, which is closely related to covariate shift and in general domain adaptation [2]. Meanwhile, causal inference has also been incorporated to improve the performance of domain adaptation [46, 45]. Most recently, the idea of representation learning is introduced to learn new features from covariates through random projections [25], informative subspace learning [7], and deep neural networks [21, 37].

## 3 Learning Balanced and Nonlinear Representations (BNR)

In this section, we first define the notations that will be used throughout this paper. Then we introduce how to convert the counterfactual prediction problem into a multi-class classification problem, and justify the rationality of this strategy. We will also present the details of how to learn nonlinear and balanced representations, and derive the closed-form solutions to the model.

**Notations.** Let $X = [X_{\mathrm{C}}, X_{\mathrm{T}}] \in \mathbb{R}^{d \times N}$ denote the covariates of all units, where $X_{\mathrm{C}} \in \mathbb{R}^{d \times N_C}$ is the control group with $N_C$ units, and $X_{\mathrm{T}} \in \mathbb{R}^{d \times N_T}$ is the treatment group with $N_T$ units. $N$ is the total number of units, and $d$ is the number of covariates for each unit. $\phi : x \in \mathbb{R}^d \to \phi(x) \in \mathbb{F}$ is a nonlinear mapping function from sample space $\mathbb{R}$ to an implicit feature space $\mathbb{F}$. $T \in \mathbb{R}^{N \times 1}$ is a binary vector to indicate if the units received treatments or not. $Y \in \mathbb{R}^{N \times 1}$ is an outcome vector. The elements in $Y$ could be either discrete or continuous values.

### 3.1 From Counterfactual Prediction to Multi-Class Classification

When estimating the treatment effects as shown in Eq.(2), we only have the observed outcome $Y_k(1)$, but need to estimate the counterfactual $\hat{Y}_k(0)$. Ideally, we would train a model $\hat{Y}_k(0) = \mathcal{F}_{cf}(x_k)$ that can predict the counterfactual for any units, given the covariate vector $x_k$. One strategy is to build a predictive model (e.g., regression) that maps each unit $x_i$ to its output $Y_i$, which has been extensively

studied before. Alternatively, we can convert the counterfactual prediction problem into a multi-class classification problem.

Given a set of units $X$ and the corresponding outcome vector $Y$, we aim to learn a predictive model $\mathcal{F}_{cf}(x_k)$ that maps from the covariate space to the outcome space. In particular, we propose to seek an intermediate representation space in which the units close to each other should have very similar outcomes. The outcome vector $Y$ usually contains continuous values. We categorize outcomes in $Y$ into multiple levels on the basis of the magnitude of outcome value, and consider them as (pseudo) class labels. Clustering or kernel density estimation can be used for discretizing $Y$. Finally, $Y$ is converted to a (pseudo) class label vector $Y_c$ with $c$ categories. For example, $Y = [0.3, 0.5, 1.1, 1.2, 2.4]$ could be categorized as $Y_3 = [1, 1, 2, 2, 3]$. As a result, we could use $Y_c$ and $X$ to train a classifier.

Note that the $Y_c$ actually contains ordinal labels, as the discretized labels carry additional information. In particular, the labels $[1, 2, 3]$ are not totally independent. We actually assume that *Class 1* should be more close to *Class 2* than *Class 3*, since the outcome values in *Class 1* are closer to those in *Class 2*. We will make use of such ordinal label information when designing the classification model.

## 3.2 Learning Nonlinear Representations via Ordinal Scatter Discrepancy

To obtain effective representations from $X$, we propose to train a nonlinear classifier in a reproducing kernel Hilbert space (RKHS). The reasons of employing the RKHS based nonlinear models are as follows. First, compared to linear models, nonlinear models are usually more capable of dealing with complicated data distributions. It is well known that the treatment and control groups might have diverse distributions, and the nonlinear models would be able to tightly couple them in a shared low-dimensional subspace. Second, the RKHS based nonlinear models usually have closed-form solutions because of the kernel trick, which is beneficial for handling large-scale data.

Let $\phi(x_i)$ denote the mapped counterpart of $x_i$ in kernel space, and then $\Phi(X) = [\phi(x_1), \phi(x_2), \cdots, \phi(x_N)]$. In light of the maximum scatter difference criterion [26], we take into account the ordinal label information, and propose a novel criterion named Ordinal Scatter Discrepancy (OSD) to achieve the desired data distribution after projecting $\Phi(X)$ to a low-dimensional subspace. In particular, OSD minimizes the within-class scatter, and meanwhile maximize the noncontiguous-class scatter matrix. Let $P$ denote a transformation matrix, OSD maps samples onto a subspace by maximizing the differences of noncontiguous-class scatter and within-class scatter. We perform OSD in kernel space to learn nonlinear representations, and have the following objective function:

$$\arg\max_{P} \quad F(P, \Phi(X), Y_c) = \mathrm{tr}(P^\top (K_I - \alpha K_W)P),$$
$$s.t. \qquad P^\top P = \mathrm{I}, \tag{3}$$

where $\alpha$ is a non-negative trade-off parameter, $\mathrm{tr}(\cdot)$ is the trace operator for matrix, and I is an identity matrix. The orthogonal constraint $P^\top P = \mathrm{I}$ is introduced to reduce the redundant information in projection.

In Eq.(3), $K_I$ and $K_W$ are the noncontiguous-class scatter matrix and within-class scatter matrix in kernel space, respectively. The detailed definitions are:

$$K_I^\Phi = \frac{c(c-1)}{2} \sum_{i=1}^{c} \sum_{j=i+1}^{c} e^{(j-i)}(m_i - m_j)(m_i - m_j)^\top \tag{4}$$

$$K_W^\Phi = \frac{1}{N} \sum_{i=1}^{c} \sum_{j=1}^{n_i} (\xi(x_{ij}) - \bar{m})(\xi(x_{ij}) - \bar{m}_i)^\top \tag{5}$$

where $\xi(x_{ij}) = [k(x_1, x_{ij}), k(x_2, x_{ij}), \cdots, k(x_N, x_{ij})]^\top$, $m_i$ is the mean vector of $\xi(x_{ij})$ that belongs to the $i$-th class, $\bar{m}$ is the mean vector of all $\xi(x_{ij})$, and $n_i$ is the number of units in the $i$-th class. $k(x_i, x_j) = \langle \phi(x_i), \phi(x_j) \rangle$ is a kernel function, which is utilized to avoid calculating the explicit form of function $\phi$ (i.e., the kernel trick).

Eq. (4) characterizes the scatter of a set of classes with (pseudo) ordinal labels. It measures the scatter of every pair of classes. The factor $e^{(j-i)}$ is used to penalize the classes that are noncontiguous. The intuition is that, for ordinal labels, we may expect the contiguous classes will be close to each other after projection, while the noncontiguous classes should be pushed away. Therefore, we put larger

weights for the noncontiguous classes. For example, $e^{(2-1)} < e^{(3-1)}$, since *Class 1* should be more close to *Class 2* than *Class 3*, as we explained in Section 3.1.

Eq. (5) measures the within-class scatter. We expect that the units having the same (pseudo) class labels will be very close to each other in the feature space, and therefore they will have similar feature representations after projection.

The differences between the proposed OSD criterion and other discriminative criteria (e.g., Fisher criterion, maximum scatter difference criterion) are two-fold. (1) OSD criterion learns nonlinear projection and feature representations in the RKHS space; (2) OSD explicitly makes use of the ordinal label information that are usually ignored by existing criteria. Moreover, the maximum scatter difference criterion is a special case of OSD.

### 3.3 Learning Balanced Representations via Maximum Mean Discrepancy

Balanced distributions of control and treatment groups, in terms of covariates, would greatly facilitate the causal inference methods such as NNM. To this end, we adopt the idea of maximum mean discrepancy (MMD) [4] when learning the transformation $P$, and finally obtain balanced nonlinear representations. The MMD criterion has been successfully applied to some problems like domain adaptation [28].

Assume that the control group $X_C$ and treatment group $X_T$ are random variable sets with distributions $\mathcal{P}$ and $\mathcal{Q}$, MMD implies the empirical estimation of the distance between $\mathcal{P}$ and $\mathcal{Q}$. In particular, MMD estimates the distance between nonlinear feature sets $\Phi(X_C)$ and $\Phi(X_T)$, which can be formulated as:

$$\text{Dist}(\Phi(X_C), \Phi(X_T)) = \| \tfrac{1}{N_C} \sum_{i=1}^{n_C} \phi(X_{Ci}) - \tfrac{1}{N_T} \sum_{i=1}^{n_T} \phi(X_{Ti}) \|_{\mathbb{F}}^2, \tag{6}$$

where $\mathbb{F}$ denotes a kernel space.

By utilizing the kernel trick, $Dist(\Phi(X_C), \Phi(X_T))$ in the original kernel space can be equivalently converted to:

$$\text{Dist}(\Phi(X_C), \Phi(X_T)) = \text{tr}(KL), \tag{7}$$

where $K = \begin{bmatrix} K_{CC} & K_{CT} \\ K_{TC} & K_{TT} \end{bmatrix}$ is a kernel matrix, $K_{CC}$, $K_{TT}$, and $K_{TC}$ are kernel matrices defined on control group, treatment group, and cross groups, respectively. $L$ is a constant matrix. If $x_i, x_j \in X_C$, $L_{ij} = \frac{1}{N_C^2}$; if $x_i, x_j \in X_T$, $L_{ij} = \frac{1}{N_T^2}$; otherwise, $L_{ij} = -\frac{1}{N_C N_T}$.

As all the units are projected into a new space via projection $P$, we need to measure the MMD for new representations $\Psi(X_C) = P^\top \Phi(X_C)$ and $\Psi(X_T) = P^\top \Phi(X_T)$, and rewrite Eq.(7) into the following form after some derivations:

$$\text{Dist}(\Psi(X_C), \Psi(X_T)) = \text{tr}(P^\top KLKP). \tag{8}$$

### 3.4 BNR Model and Solutions

The representation learning objectives described in Section 3.2 and Section 3.3 are actually performed on the same data set with different partitions. For nonlinear representation learning, we merge the control group and treatment group, assign a (pseudo) ordinal label for each unit, and then learn discriminative nonlinear features accordingly. For balanced representation learning, we aim to mitigate the distribution discrepancy between control group and treatment group. Two learning objectives are motivated from different perspectives, and therefore they are complementary to each other. By combing the objectives for nonlinear and balanced representations in Eq.(3) and Eq.(8), we can extract effective representations for the purpose of treatment effect estimation.

The objective function of BNR is formulated as follows:

$$\begin{aligned} \arg\max_{P} \quad & F(P, \Phi(X), Y_c) - \beta \text{Dist}(\Psi(X_C), \Psi(X_T)) \\ & = \text{tr}(P^\top (K_I - \alpha K_W) P) - \beta \text{tr}(P^\top KLKP), \\ s.t. \quad & P^\top P = \mathrm{I}, \end{aligned} \tag{9}$$

where $\beta$ is a trade-off parameter to balance the effects of two terms. A negative sign is added before $\beta \text{Dist}(\Psi(X_C), \Psi(X_T))$ in order to adapt it into this maximization problem.

The problem Eq.(9) can be efficiently solved by using a closed-form solution described in Proposition 1. The proof is provided in the supplementary document due to space limit.

**Proposition 1.** *The optimal solution of $P$ in problem Eq.(9) is the eigenvectors of matrix $(K_I - \alpha K_W - \beta KLK)$, which correspond to the $m$ leading eigenvalues.*

## 4 BNR for Nearest Neighbor Matching

Leveraging on the balanced nonlinear representations extracted from observational data, we propose a novel nearest neighbor matching estimator named BNR-NNM.

After obtaining the transformation $P$ in kernel space, we could generate nonlinear and balanced representations for control and treated units as: $\hat{X}_C = P^\top K_C$, $\hat{X}_T = P^\top K_T$, where $K_C$ and $K_T$ are kernel matrices defined in control and treatment groups, respectively. Then we follow the basic idea of nearest neighbor matching. On the new representations $\hat{X}_C$ and $\hat{X}_T$, we calculate the distance between each treated unit and control unit, and choose the one with the smallest distance. The outcome of the selected control unit serves as the estimation of counterfactual. Finally, the average treatment effect on treated (ATT) can be calculated, as defined in Eq.(2). The complete procedures of BNR-NNM are summarized in Algorithm 1.

The estimated ATT is dependent on the transformation matrix $P$. Although $P$ is optimal for the representation learning model Eq.(9), it might not be optimal for the whole causal inference process, for three reasons. First, the model Eq.(9) contains two major hyperparameters, $\alpha$ and $\beta$. Different "optimal" transformations $P$ would be obtained with different parameter settings. Second, the ground-truth label information required by supervised learning are unknown. Recall that we categorize the outcome vector as pseudo labels, which introduces considerable uncertainty. Third, the ground-truth information of causal effect is unknown in observational studies with real-world data. Therefore, it is impossible to use the faithful supervision information of causal effect to guide the learning process. These uncertainties from three perspectives might result in an unreliable estimation of ATT.

---
**Algorithm 1.** *BNR-NNM*
---
**Input:** Treatment group $X_T \in \mathbb{R}^{d \times N_t}$
        Control group $X_C \in \mathbb{R}^{d \times N_c}$
        Outcome vectors $Y_T$ and $Y_C$
        Total sample size $N$
        Kernel function $k$
        Parameters $\alpha$, $\beta$, $c$
1: Convert outcomes to (pseudo) ordinal labels
2: Construct $K_I$ and $K_W$ using Eqs.(4) and (5)
3: Construct kernel matrix $K$ using Eq.(7)
4: Learn the transformation $P$ using Eq.(9)
5: Construct kernel matrix $K_C$ and $K_T$
6: Project $K_C$ and $K_T$ using $P$
    $\hat{X}_C = P^\top K_C$, $\hat{X}_T = P^\top K_T$.
7: Perform NNM between $\hat{X}_C$ and $\hat{X}_T$
8: Estimate the ATT $A$ from Eq.(2)
**Output:** Return $A$
---

Thus, we present two strategies to tackle the above issue. (1) *Median causal effect from multiple estimations.* Following the randomized NNM estimator [25], we implement multiple settings of BNR-NNM with different parameters $\alpha$, $\beta$ and $c$, calculate multiple ATT values, and finally choose the median value as the final estimation. In this way, a robust estimation of causal effect can be obtained. (2) *Model selection by cross-validation.* Alternatively, the cross-validation strategy can be employed to select proper values for $\alpha$ and $\beta$, by equally dividing the data and pseudo labels into $k$ subsets. Although the multiple runs in the above strategies would increase the computational cost, our method is still efficient for three reasons. First, the dimension of covariates will be significantly reduced, which enables a faster matching process. Second, owing to the closed-form solution to $P$ introduced in Proposition 1, the representation learning procedure is efficient. Third, these settings are independent from each other, and therefore they can be executed in parallel.

## 5 Experiments and Analysis

**Synthetic Dataset.** *Data Generation.* We generate a synthetic dataset by following the protocols described in [41, 25]. In particular, the sample size $N$ is set to 1000, and the number of covariates $d$ is set to 100. The following basis functions are adopted in the data generation process: $g_1(x) = x - 0.5$, $g_2(x) = (x - 0.5)^2 + 2$, $g_3(x) = x^2 - 1/3$, $g_4(x) = -2\sin(2x)$, $g_5(x) = e^{-x} - e^{-1} - 1$, $g_6(x) = e^{-x}$, $g_7(x) = x^2$, $g_8(x) = x$, $g_9(x) = \mathcal{I}_{x>0}$, and $g_{10}(x) = \cos(x)$. For each unit, the covariates $x_1, x_2, \cdots, x_d$ are drawn independently from the standard normal distribution $\mathcal{N}(0, 1)$.

We only consider binary treatment in this paper, and define the treatment vector $T$ as $T|x = 1$ if $\sum_{k=1}^{5} g_k(x_k) > 0$ and $T|x = 0$ otherwise. Given covariate vector $x$ and the treatment vector $T$, the outcome variables in $Y$ are generated from the following model: $Y|x, T \sim \mathcal{N}(\sum_{j=1}^{5} g_{j+5}(x_j) + $

$T, 1$). It is obvious that $Y$ contains continuous values. The first five covariates are correlated to the treatments in $T$ and the outcomes in $Y$, simulating a confounding effect, while the rest are noisy components. By definition, the true causal effect (i.e., the ground truth of ATT) in this dataset is 1.

*Baselines and Settings.* We compare our matching estimator BNR-NNM with the following baseline methods: Euclidean distance based NNM (Eud-NNM), Mahalanobis distance based NNM (Mah-NNM) [34], PSM [31], principal component analysis based NNM (PCA-NNM), locality preserving projections based NNM (LPP-NNM), and randomized NNM (RNNM) [25].

PSM is a classical causal inference approach, which estimates the propensity scores for each control or treated unit using logistic regression, and then perform matching on these scores. As our approach learns new representations via transformations, we also implement two matching estimators based on the popular subspace learning methods PCA [22] and LPP [13]. The nearest neighbor matching is performed on the low-dimensional feature space learned by PCA and LPP, respectively. RNNM is the state-of-the-art matching estimator, especially for high-dimensional data. It projects units to multiple random subspaces, performs matching in each of them, and finally selects the median value of estimations. In RNNM, the number of random projections is set to 20. The proposed BNR-NNM and RNNM share a similar idea on projecting data to low-dimensional subspaces, but they have different motivations and learn different data representations.

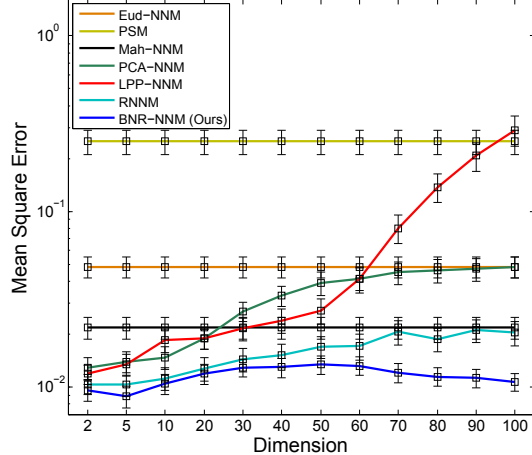

Figure 1: MSE of different estimators on the synthetic dataset. Note that Eud-NNM and Mah-NNM only involve matching in the original 100 dimensional data space.

The major parameters in BNR-NNM include $\alpha$, $\beta$, and $c$. In the experiments, $\alpha$ is empirically set to 1. $\beta$ is chosen from $\{10^{-3}, 10^{-1}, 1, 10, 10^3\}$. The number of categories $c$ is chosen from $\{2, 4, 6, 8\}$. As described in Section 4, the median ATT of multiple estimations is used as the final result. We use the Gaussian kernel function $k(x_i, x_j) = \exp(-\|x_i - x_j\|^2 / 2\sigma^2)$, in which the bandwidth parameter $\sigma$ is empirically set to 5. In the experiments we observe that our approach allows flexible settings for these parameters, and intuitively selecting parameters from a wider range would lead to a robust estimation of ATT.

*Results and Discussions.* To ensure a robust estimation of the performance of each matching estimator, we repeat the data generation process 500 times, calculate the ATT for each estimator in every replication, and compute the mean square error (MSE) with standard error (SD) for each estimator over all of the replications. Eud-NNM and Mah-NNM perform matching in the original covariate space, and PSM maps each unit to a single score. Thus we only have a single point estimation for each of them. For PCA-NNM, LPP-NNM, RNNM and our method, we can choose the dimension of feature space where the matching is conducted. Specifically, we increase the dimension from 2 to 100, and calculate MSE and SD in each case. Figure 1 shows the MSE and SD (shown as error bars) of each estimator when varying the dimensions. We observe from Figure 1 that the proposed estimator BNR-NNM obtains lower MSE than all other methods in every case. The lowest MSE is achieved when the dimension is 5. In addition, we have analyzed the sensitivity of parameter settings. The detailed results are provided in the supplementary document.

**IHDP Dataset with Simulated Outcomes.** IHDP data [16] is an experimental dataset collected by the Infant Health and Development Program. In particular, a randomized experiment was conducted, where intensive high-quality care were provided to the low-birth-weight and premature infants. By using the original data, an observation study can be conducted by removing a nonrandom subset of the treatment group: all children with non-white mothers. After this preprocessing step, there are in total 24 pretreatment covariates (excluding race) and 747 units, including 608 control units and 139 treatment units. The outcomes are simulated by using the pretreatment covariates and the treatment assignment information, in order to hold the unconfoundedness assumption.

Due to the space limit, the outcome simulation procedures are provided in the supplementary document. We repeat such procedures for 200 times and generate 200 sets of simulated outcomes, in order to conduct extensive evaluations. For each set of simulated outcomes, we run our method and the baselines introduced above, and report the results in Table 1. We use the error in average treatment effect on treated (ATT), $\varepsilon_{ATT}$, as the evaluation metric. It is defined as the absolute difference between true ATT and estimated ATT ($\widehat{ATT}$), i.e., $\varepsilon_{ATT} = |ATT - \widehat{ATT}|$. Table 1 shows that the proposed BNR-NNM estimator outperforms most baselines, which further validates the effectiveness of the balanced and nonlinear representations.

Table 1: Results on IHDP dataset.

| Method | $\varepsilon_{ATT}$ |
|---|---|
| Eu-NNM | 0.18±0.06 |
| Mah-NNM | 0.31±0.12 |
| PSM | 0.26±0.08 |
| PCA-NNM | 0.19±0.11 |
| LPP-NNM | 0.25±0.13 |
| RNNM | **0.16±0.07** |
| BNR-NNM | **0.16±0.06** |

**LaLonde Dataset with Real Outcomes.** The LaLonde dataset is a widely used benchmark for observational studies [23]. It consists of a treatment group and a control group. The treatment group contains 297 units from a randomized study of a job training program (the "National Supported Work Demonstration"), where an unbiased estimate of the average treatment effect is available. The original LaLonde dataset contains 425 control units that are collected from the Current Population Survey. Recently, Imai *et al.* augmented the data by including 2,490 units from the Panel Study of Income Dynamics [18]. Thus, the sample size of control group is increased to 2,915. For each sample, the covariates include age, education, race (black, white, or Hispanic), marriage status, high school degree, earnings in 1974, and earnings in 1975. The outcome variable is earnings in 1978. In this benchmark dataset, the unbiased estimation of ATT is $886 with a standard error of $448.

We compare our estimator with the baselines used in the previous experiments. In addition, we also compare with a recently proposed matching estimator, covariate balancing propensity score (CBPS) [18] and a deep neural network (DNN) method [37]. CBPS aims to achieve balanced distributions between control and treatment groups by adjusting the weights for covariates. The DNN method utilizes a deep neural network architecture for counterfactual regression, which is the state-of-the-art method on representation learning based counterfactual inference. For BNR-NNM, we use the same settings for $\beta$ and $c$ as in the previous experiments.

Table 2: Results on LaLonde dataset. BIAS (%) is the bias in percentage of the true effect.

| Method | ATT | SD | BIAS (%) |
|---|---|---|---|
| Ground Truth | 886 | 488 | N/A |
| Eu-NNM | -565.9 | 592.8 | 164% |
| Mah-NNM | -67.9 | 526.1 | 108% |
| PSM | -947.6 | 567.9 | 201% |
| PCA-NNM | -499.8 | 592.5 | 156% |
| LPP-NNM | -457.1 | 581.2 | 152% |
| RNNM | -557.6 | 584.9 | 163% |
| CBPS | 423.3 | 1295.2 | 52% |
| DNN | 742.0 | N/A | 16% |
| BNR-NNM | 783.6 | 546.3 | **12%** |

Table 2 shows the ground truth of ATT, and the estimations of different methods. We can observe from Table 2 that CBPS and DNN obtain better results than other baselines, as both of them consider the balanced property across treatment and control groups. Moreover, our BNR-NNM estimator achieves the best result, due to the fully exploitation of balanced and nonlinear feature representations. The evaluations on runtime behavior of each compared method are provided in the supplementary document due to space limit.

## 6 Conclusions

In this paper, we propose a novel matching estimator based on balanced and nonlinear representations for treatment effect estimation. Our method leverages on the predictive power of machine learning models to estimate counterfactuals, and achieves balanced distributions in an intermediate feature space. In particular, an ordinal scatter discrepancy criterion is designed to extract discriminative features from observational data with ordinal pseudo labels, while a maximum mean discrepancy criterion is incorporated to achieve balanced distributions. Extensive experimental results on three synthetic and real-world datasets show that our approach provides more accurate estimation of causal effects than the state-of-the-art matching estimators and representation learning methods. In future work, we will extend the balanced representation learning model to other causal inference strategies such as weighting and regression, and design estimators for multiple levels of treatments.

**Acknowledgement.** This research is supported in part by the NSF IIS award 1651902, ONR Young Investigator Award N00014-14-1-0484, and U.S. Army Research Office Award W911NF-17-1-0367.

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
