[Supplementary Material]

# Matching on Balanced Nonlinear Representations for Treatment Effects Estimation

## Supplementary Document

**Sheng Li**
Adobe Research
San Jose, CA
sheli@adobe.com

**Yun Fu**
Northeastern University
Boston, MA
yunfu@ece.neu.edu

## 1  Proof of Proposition 1

The objective function of the proposed balanced and nonlinear representations (BNR) model is:

$$
\begin{aligned}
\arg\max_{P} \quad & F(P, \Phi(X), Y_c) - \beta\mathrm{Dist}(\Psi(X_C), \Psi(X_T)) \\
&= \mathrm{tr}(P^\top(\alpha K_W - K_I)P) - \beta\mathrm{tr}(P^\top KLKP), \\
s.t. \quad & P^\top P = \mathrm{I},
\end{aligned}
\tag{1}
$$

where $\beta$ is a trade-off parameter to balance the effects of two terms. A negative sign is added before $\beta\mathrm{Dist}(\Psi(X_C), \Psi(X_T))$ in order to adapt it into this maximization problem.

The problem Eq.(1) can be efficiently solved by using a closed-form solution described in Proposition 1.

**Proposition 1** *The optimal solution of $P$ in problem Eq.(1) is the eigenvectors of matrix $(\alpha K_I - K_W - \beta KLK)$, which correspond to the $m$ leading eigenvalues.*

**Proof.** The Lagrangian function of Eq.(1) is:

$$
\mathcal{L} = \mathrm{tr}(P^\top(\alpha K_I - K_W - \beta KLK)P) - \mathrm{tr}((P^\top P - \mathrm{I})Z),
\tag{2}
$$

where $Z$ is a Lagrangian multiplier.

By setting the derivative of Eq.(2) w.r.t. $P$ to zero, we have:

$$
\frac{\partial \mathcal{L}}{\partial P} = (\alpha K_I - K_W - \beta KLK)P = PZ.
\tag{3}
$$

Eq.(3) is a standard eigen-decomposition problem. Therefore, the optimal solution of $P$ in Eq.(1) is the eigenvectors of matrix $(\alpha K_I - K_W - \beta KLK)$ corresponding to the $m$ leading eigenvalues. □

## 2  Experimental Settings and Additional Results

### 2.1  Additional Results on Synthetic Dataset

To illustrate the sensitivity of parameter settings, Figure 1 (a) and (b) show the ATT with different values of $\beta$ and different number of categories $c$, respectively, when the dimension is increased from 2 to 20. The ground truth of ATT is 1. We can observe that most of the estimations are quite close to 1 (shown as dashed lines), and therefore the median value of those estimations will be close to 1 as well. These results demonstrate that the proposed BNR-NNM estimator is able to provide a robust estimation of causal effect.

(a) ATT with different values of $\beta$, when $c = 4$ and $\alpha = 1$.

(b) ATT with different number of categories ($c$), when $\beta = 1$ and $\alpha = 1$.

Figure 1: ATT estimated by our approach using different settings on synthetic dataset. The ground truth of ATT is 1.

## 2.2 Outcome Simulation Procedures on IHDP Dataset

Given the covariate matrix $X$ and treatment indicator vector $T$, we follow the procedures suggested by Hill [1] to simulate the outcomes:

- $Y(0) = \exp((X + W)\beta) + Z_0$, where $W$ is an offset matrix with every element equal to 0.5; $\beta \in \mathbb{R}^{d \times 1}$ is a vector of regression coefficients $(0, 0.1, 0.2, 0.3, 0.4)$ randomly sampled with probabilities $(0.6, 0.1, 0.1, 0.1, 0.1)$; $Z_0 \in \mathbb{R}^{n \times 1}$ is a vector of elements randomly sampled from the standard normal distribution $N(0, 1)$.

- $Y(1) = X\beta - \omega + Z_1$, where $\beta$ follows the same definition as described above. $\omega \in \mathbb{R}^{n \times 1}$ is a vector with every element to some constant that makes ATT equal to 4. Similar to $Z_0$, $Z_1 \in \mathbb{R}^{n \times 1}$ is also a vector of elements randomly drawn from the standard normal distribution $N(0, 1)$.

- The factual outcome vector is defined as $Y^F = Y(1) \odot T + Y(0) \odot (1 - T)$ and the counterfactual outcome vector $Y^{CF} = Y(1) \odot (1 - T) + Y(0)^T$, where $\odot$ represents the element-wise product.

## 2.3 Evaluation on Efficiency

Although the proposed BNR-NNM involves a representation learning process and model selection procedures, it is still efficient compared with the existing matching estimators. The efficiency of BNR-NNM leverages on the following factors: (1) matching in low-dimensional representation space

Table 1: Computing time (in seconds) of different estimators on synthetic dataset.

| Method | Time (seconds) |
|---|---|
| Eu-NNM | 0.07 |
| Mah-NNM | 1.79 |
| PSM | 0.27 |
| PCA-NNM | 0.04 |
| LPP-NNM | 0.25 |
| RNNM | 0.02 |
| BNR-NNM (Ours) | 0.35 |

is much faster than in the original high-dimensional covariate space; (2) BNR has a closed-form solution; (3) multiple parameter settings can be executed in parallel. Moreover, we empirically evaluate the runtime behavior of BNR-NNM and other baselines on the synthetic dataset. The sample size is 1000 and the dimension of covariates is 100. For PCA-NNM, LPP-NNM, RNNM, and our method, we reduce the dimension of covariates from 100 to 5. Table 1 shows the computing time of different estimators. We can observe that the time cost of our estimator is comparable with that of other baselines.

# References

[1] Jennifer L Hill. Bayesian nonparametric modeling for causal inference. *Journal of Computational and Graphical Statistics*, 20(1):217–240, 2012.