[Reviews · NeurIPS 2017]

Reviewer 1



This paper presents a novel mechanism for estimating treatment effects that is robust to imbalanced treatment and control groups, as well as circumvents issues with bias in high dimensional data. This is achieved by learning balanced and non-linear representations of observational data that are of lower dimension which is used in conjunction with nearest neighbor matching. The proposed method shows improvements over traditional techniques on both real-world and synthetic data. Overall this paper is clearly written. The authors adequately motivate the purpose of the paper, which is to improve on the shortcomings in current treatment effects estimation methods. The proposed BNR method is explained in adequate detail, both mathematically and through pseudo code. Experimental settings required for reproducibility are provided in supplementary material. I believe the paper would be improved by additional details from the authors regarding potential weaknesses or threats to validity, eg. are there use cases when the proposed technique would not work well? The conclusion section of the paper adds very little. It could likely be cut and the space better used in the experiments and analysis section where the authors must be very brief due to space constraints.

Reviewer 2



The paper proposes a new matching algorithm for estimation of treatment effects in observational studies. The algorithm is appropriate when the number of covariates is large. It finds a projection of the original covariate space into a lower dimensional one and applies a standard matching approach such as nearest neighbor matching in this new space. The proposed low dimensional nonlinear projection has two objectives: (1) maximize separation between outcomes and (2) maximize the overlap between cases and controls. The new algorithm is evaluated on one synthetic and two real data sets. The results indicate the proposed method is better than several alternatives. Positives: + the proposed objective function is intuitively appealing + the experiments were well designed and the results are positive + the paper is easy to read Negatives: - the proposed ordinal scatter discrepancy criterion is a heuristic that is sounds reasonable. However, it would require a deeper experimental or theoretical study to truly understand its advantages and drawbacks. - Any type of causal study on the observational data needs to come with caveats and understanding of potential pitfalls. This paper is not discussing it and presents the algorithm only in a positive light. It would be useful for the paper to at least have a "buyer beware" discussion. - There are several moving parts required to make the proposed algorithm work, such as discretizing the outcomes, selecting the kernel, deciding on the dimensionality, selecting the hyperparameters. It would be important to see how sensitive the algorithm is to those choices. From the presented results, it is fair to ask if the shown results might be due to some overfitting - the paper will need to be carefully proofread -- there are numerous grammar mistakes throughout.

Reviewer 3



The paper addresses the problem of finding matched control set for counterfactual prediction. This is done by nearest neighbor matching but in a lower dimensional space. The nonlinear lower dimensional projection (P) is found by maximizing the difference between class-scatter and within-class-scatter, and additionally minimizing the difference between control and treatment group covariate distributions. The experiment shows that the proposed method performs better than existing approaches in a cross-validation set-up. The authors suggest converting real outcomes to ordinal values. Is there a formal approach of how this conversion should be done, e.g., how many ordinal levels? Can the authors also present results for real valued outcomes without converting them to ordinals?

Reviewer 4



The authors present a kernel learning model for use in causal analysis. The model is optimized to learn a representation that minimizes maximum-mean-discrepancy (MMD) while maximizing a novel criterion dubbed Ordinal Scatter Discrepancy (OSD). MMD represents an existing metric for discriminating between distributions, and OSD measures the representation’s between-class scattering less its within-class scattering (where classes are discrete treatment outcomes - additional adjustments are described for ordinal classes). The learned representation is then applied in nearest-neighbor-matching for causal analysis. The paper is well written and offers both interesting algorithmic results as well as empirical validation against a popular benchmark dataset yielding lower bias than all baselines including a DNN method representing the current state-of-the-art.